# A cryo-electron microscopic approach to elucidate protein structures from human brain microsomes

Marios L Tringides, Zhemin Zhang, Christopher E Morgan, Chih-Chia Su , Edward W Yu

**We recently developed a "Build and Retrieve" cryo-electron microscopy (cryo-EM) methodology, which is capable of simultaneously producing near-atomic resolution cryo-EM maps for several individual proteins from a heterogeneous, multiprotein sample. Here we report the use of "Build and Retrieve" to define the composition of a raw human brain microsomal lysate. From this sample, we simultaneously identify and solve cryo-EM structures of five different brain enzymes whose functions affect neurotransmitter recycling, iron metabolism, glycolysis, axonal development, energy homeostasis, and retinoic acid biosynthesis. Interestingly, malfunction of these important proteins has been directly linked to several neurodegenerative disorders, such as Alzheimer's, Huntington's, and Parkinson's diseases. Our work underscores the importance of cryo-EM in facilitating tissue and organ proteomics at the atomic level.**

## Introduction

Recently, the rapid progress of mass spectrometry has opened up new horizons for the field of systems biology as it has allowed us to begin to elucidate the proteomes of different human tissues and organs. This exciting omics approach has resulted in very large–scale human proteome programs, including the Human Proteome Project (Paik et al, 2012), the Human Protein Atlas program (Uhlen et al, 2010), and the human tissue proteome map (Uhlen et al, 2015), to facilitate the quantification of spatially localized proteins and biomolecules down to the single cell level. The integrated systems approach for studying tissues and organs has previously been thought to be unavailable in the field of X-ray crystallography and cryo-electron microscopy (cryo-EM), where these structural biology techniques typically require homogeneous and pure samples. As a result, these tools have had difficulties extending to the systems biology approach to investigate specimens possessing a mixture of proteins and biomacromolecules in a complex, heterogeneous environment.

To address this challenge and originate an approach of using cryo-EM to study tissue and organ samples, we recently developed the "Build and Retrieve" (BaR) cryo-EM methodology (Su et al, 2021). BaR is an iterative methodology capable of performing in silico purification and sorting of images from a large, heterogeneous dataset consisting of numerous diverse proteins and biomacromolecules. It is capable of deconvoluting images and allowing for the simultaneous production of near-atomic resolution cryo-EM maps of individual proteins from a heterogeneous, multiprotein sample.

To exemplify the ability of BaR in elucidating systems structural proteomics of human tissues, this technique combined with cryo-EM allowed for rapid identification and structural determination of proteins within human brain microsomes. The brain is a fascinating system in the human body. It is one of the most divergent, yet specialized organs (Kim et al, 2014; Wilhelm et al, 2014), possessing the second highest number of tissue-specific enriched genes. Transcriptome analysis indicates that 82% of all human proteins are expressed in the brain (Uhlen et al, 2015). Of particular importance is the prevalence of many neurodegenerative disorders, such as Alzheimer's, Huntington's, and Parkinson's diseases. It is also commonly known that the aged brain is much more susceptible to these pathologies. To date, ~5.8 million Americans suffer from Alzheimer's disease (CDC, 2022); 1.2 million from Parkinson's (Marras et al, 2018); 400,000 from multiple sclerosis (Dilokthornsakul et al, 2016); 30,000 from amyotrophic lateral sclerosis; and 30,000 from Huntington's disease. Many of these conditions are caused or exacerbated, in part, by proteins becoming misfolded, aggregated, and/or functionally damaged. Therefore, the structural information of these proteins in their native state is critical for future palliative and curative treatment development.

In this study, we enriched proteins from raw lysate of human brain microsomes using size-exclusion chromatography. We then used single-particle cryo-EM imaging to simultaneously identify and solve cryo-EM structures of five different brain enzymes with functions important for neurotransmitter recycling, iron metabolism, glycolysis, axonal development, energy homeostasis, and retinoic acid biosynthesis. Importantly, these essential enzymes are linked to several neurodegenerative disorders, such as Alzheimer's, Huntington's, and Parkinson's diseases, suggesting that they may play critical roles in their pathogenesis.

---

Department of Pharmacology, Case Western Reserve University School of Medicine, Cleveland, OH, USA

Correspondence: edward.w.yu@case.edu

# Results

Enrichment of the brain microsomal raw lysate using size-exclusion chromatography resulted in two major peaks of proteins. The sizes of these two peaks are located at 100–300 kD and 300–500 kD (Fig S1). We collected single-particle cryo-EM images of the respective peaks and then processed the data using the BaR methodology (Su et al, 2021). Briefly, several iterative rounds of 2D classification permitted us to sort the images into different protein classes (Figs S2 and S3). Particle subsets with similar structural features are combined and then sampled using 3D class ab initio reconstruction. Classes with clear high-resolution features are selected for further processing. Although these initial maps have some clear features, they often lack complete views. Therefore, these initial maps are used as templates for a series of heterogeneous 3D classifications to extract additional particle views from the cleaned initial particle stack. Updated particle stacks are then processed using multiple rounds of 2D and 3D classifications before cryo-EM map construction and final structural refinement.

Collectively, the BaR approach was then used to reveal enzyme identities and solve cryo-EM structures of these brain enzymes with resolutions ranging between 2.69 and 3.40 Å (Table S1). These enzymes were identified as glutamine synthetase (GS), ferritin (FT), dihydropyrimidinase-related protein 2 (DPYSL2), glyceraldehyde 3-phosphate dehydrogenase (GAPDH), and retinaldehyde dehydrogenase 1 (ALDH1A1).

We also used proteomic analysis to elucidate the composition of both of the human brain microsomal lysate peaks. We found that each lysate peak contains more than 400 proteins. A sampling of some of the most abundant proteins is listed in Table S2. The existence of the five enzymes identified by BaR in our sample was also confirmed by this proteomic analysis (Table S2).

## GS

GS is a glutamine synthesizing enzyme engaged in the recycling of synaptically released glutamate and γ-aminobutyric acid (GABA) and the detoxification of ammonia. In the brain, GS is involved in several important biological processes. It participates in the metabolic regulation of glutamate, assimilation of ammonia, recyclization of neurotransmitters, and termination of neurotransmitter signals (Liaw et al, 1995; Suárez et al, 2002). Interestingly, a strong relationship between GS and Alzheimer's plaque formation has been observed, where a high concentration of GS was found in the cerebral spinal fluid of Alzheimer's patients (Gunnersen & Haley, 1992). Studies also suggested that GS deficiency in discrete areas of the brain may play a crucial role in the pathogenesis of several other neurological disorders, including Alzheimer's disease, schizophrenia, depression, and epilepsy (Sandhu et al, 2021).

We were able to obtain a total of 27,886 projections for this class of enzyme (Fig S4 and Table S1). Based on these projections, the BaR methodology allowed us to construct a high-resolution cryo-EM map. Subsequently, we were able to identify this protein as the GS enzyme and resolve its structure to a resolution of 2.73 Å (Figs 1 and S4 and Table S1).

Human GS assembles as an oligomer of 10 subunits, creating a dimer of pentamers (Fig 1A and B). Its overall structure has a very similar fold to that of canine GS (PDB ID: 2UU7) (Krajewski et al, 2008). The two pentamers stack against each other to form two layers. Each subunit of GS consists of a small N-terminal domain (residues 2–102) and a large C-terminal domain (residues 114–372). The N-terminus of the small N-terminal domain creates an α-helix, which is connected to a β grasp of five β-strands via a random loop. The large C-terminal domain possesses nine α-helices and 11 β-strands. Based on the structure, the N-terminal domain contributes to subunit–subunit interactions within each layer of the pentamer, whereas the C-terminal domain participates in forming the catalytic site for substrate binding (Fig 1C). However, the loop residues (residues 149–157) of the C-terminal domain of each GS subunit are responsible for securing interactions between two pentameric layers.

Within the active site of the C-terminal catalytic domain of each subunit of human GS, we observed a bound $Mn^{2+}$ ion, which is anchored by residues E134, H253, and E338. This catalytic site also contains several other positively charged residues, including R299, R319, R324, R340, and R341, that contribute to substrate binding (Fig 1D). It has been observed that many of these residues are responsible for binding ADP and phosphate in the human GS enzyme (Krajewski et al, 2008).

Interestingly, there is a report on two cases of congenital human GS deficiency, which resulted in severe brain malformations with multiorgan failure and neonatal death (Häberle et al, 2005). Each infant described had a homozygous mutation in the *GS* gene, where residues R324 and R341 of the GS protein were replaced by cysteines. It appears that these two mutations are associated with reduced GS activity (Häberle et al, 2005).

## FT

FT is an iron storage protein (Honarmand Ebrahimi et al, 2015) characterized by its unique α-helical architecture designed to uptake iron. FT is widely recognized as a critical protein for iron metabolism, particularly in the brain (Fisher et al, 2007; Loh et al, 2009). It participates in providing iron availability for cellular demand and is also responsible for protecting cells against damage from iron-mediated free radicals (Harrison & Arosio, 1996). Interestingly, it has been observed that apo-FT, the iron-free form of FT, is able to convert its structure from a hollow, globular fold into an unusual architecture similar to those amyloid fibrils found in neuropathological disorders such as Alzheimer's and Parkinson's diseases (Jurado et al, 2019). Therefore, the misfolding of apo-FT may be linked to the progression of neurodegenerative disorders. It is worth noting that FT expression and the amount of $Fe^{3+}$ storage have been reported to be strongly altered in patients with Alzheimer's and Parkinson's diseases and AIDS (Connor et al, 1995; Drakesmith et al, 2005), suggesting that the expression level of this enzyme in the body could serve as a strong biomarker for a variety of pathologies.

We collected a total of 12,220 single-particle projections for this class of protein. The BaR protocol allowed us to identify this protein as the FT enzyme. We then determined the structure of this enzyme to a resolution of 2.69 Å (Figs 2 and S5 and Table S1).

Of the 183 amino acids of human FT, 173 residues are included in our final structural model. Human FT assembles as a 24-subunit oligomer, creating an overall spherical structure (Fig 2A and B). Each subunit of FT presents an all α-helical structure and forms an

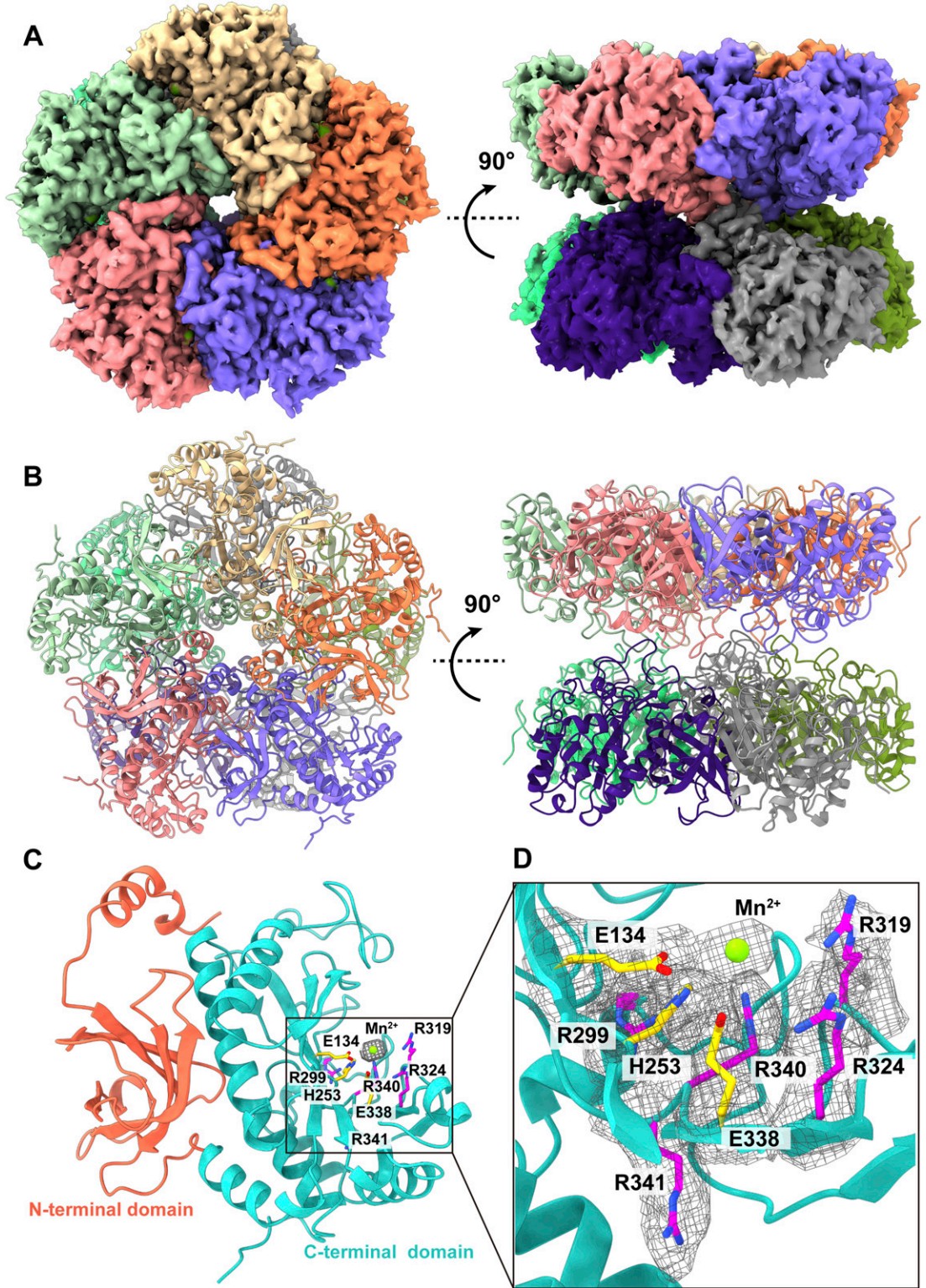

**Figure 1. Cryo-EM structure of human GS.**
**(A)** Cryo-EM maps of human GS. The 10 subunits are colored differently. **(B)** Ribbon diagrams of the structure of human GS determined by cryo-EM. GS forms a complex of D5 symmetry with two pentameric rings stacked in a two-ring conformation. In (A, B), subunits are distinguished through individual colors and match accordingly. **(C)** Structure of a subunit of GS. GS is organized into a small N-terminal domain (orange) and a large C-terminal domain (cyan). The N-terminal domain is responsible for inter-subunit interactions, whereas the C-terminal domain forms the catalytic site. Residues that participate in forming the catalytic site are in sticks. Residues (E134, H253, and E338) involved in binding $Mn^{2+}$ ion (green) are colored yellow. The cryo-EM density for bound $Mn^{2+}$ is in gray meshes. **(D)** The substrate-binding site. Residues

elongated four $\alpha$-helical bundle and a short C-terminal $\alpha$-helix (Fig 2C). In agreement with the previously determined X-ray structure (PDB ID: 2FHA) (Pozzi et al, 2015), each FT subunit contains an $Fe^{3+}$ binding site. The bound $Fe^{3+}$ binds in the middle of the four helical bundle, intimately interacting with residues E28, E63, and H66 of FT (Fig 2C and D).

Interestingly, mutagenesis studies indicate that the E63 and H66 residues play a key role in the process of rapid iron uptake in both the human and mouse FT enzymes (Lawson et al, 1989; Rucker et al, 1996). Our structure indeed depicts that these residues are critical for FT–iron interactions.

## DPYSL2

DPYSL2 plays a crucial role in the pathogenesis of the serious mental illness schizophrenia (Edgar et al, 2000; Johnston-Wilson et al, 2000; Martins-de-Souza et al, 2009). DPYSL2 may also be a key player in the development of Alzheimer's disease (Xiong et al, 2020). This enzyme is notable for its contribution to axonal growth cone collapse. It is able to mediate axonal outgrowth in the developing brain and regulate neuronal polarity to maintain proper cytoskeletal dynamics and vesicle trafficking (Hensley et al, 2011). Recently, it has been observed that DPYSL2 is capable of promoting neural stem cell differentiation into neurons, astrocytes, and oligodendrocytes that could then be used to replace necrotic cells caused by brain or spinal cord injuries (Xiong et al, 2020), suggesting a potential role for this enzyme in contributing to the development of neural stem cell–based therapeutic treatments for brain and spinal cord injuries.

We detected a total of 109,192 single-particle images for this protein. Based on the cryo-EM map, we identified that this protein is the DPYSL2 enzyme. We then determined the cryo-EM structure of this enzyme to a resolution of 2.76 Å (Figs 3 and S6 and Table S1).

The human DPYSL2 enzyme is tetrameric in oligomerization (Fig 3A and B). Each subunit of DPYSL2 consists of 572 amino acids. Residues 14–506 of each subunit are included in the structural model. Our cryo-EM structure of DPYSL2 is in good agreement with crystal structures of murine CRMP1 (PDB ID: 1KCX) (Deo et al, 2004) and human CRMP2 (PDB ID: 5MKV) (Stenmark et al, 2007). Each DPYSL2 molecule consists of a small N-terminal $\beta$-domain (residues 14–68), which possesses seven $\beta$-strands, and a large $\alpha/\beta$ TIM-barrel domain (residues 69–487), containing 19 $\alpha$-helices and 10 $\beta$-strands (Fig 3C). The C-terminus of human DPYSL2 forms an extended unstructured random segment (residues 488–506) (Fig 3C). It appears that this C-terminal tail has no sequence homology to other proteins (Stenmark et al, 2007). In addition, the C-terminal tail has been characterized as a site of proteolytic cleavage (Deo et al, 2004). Unexpectedly, we observed relatively strong cryo-EM densities for a portion of this C-terminus tail (residues 488–496). The structure indicates that each subunit of this C-terminal tail (residues 488–496) appears to cross over to the next subunit, directly contacting the next subunit to enhance subunit–subunit interactions (Fig 3A and B).

The large $\alpha/\beta$ TIM-barrel domain possesses a substrate-binding site. Although our structure lacks a bound substrate, it has been previously observed that residues D80, Q91, E111, Y167, and R173 are engaged in substrate binding (Moutal et al, 2019). Interestingly, a missense mutant Q91R of DPYSL2 has been found to cause seizures. Moreover, lacosamide, an anti-seizure drug, has been found to be effective and significantly reduce seizure frequency in patients with uncontrolled partial-onset seizures, where this anti-seizure drug is bound by residues Q91, E360, S363, K418, I420, and P443 of this enzyme (Ben-Menachem et al, 2007; Moutal et al, 2019) (Fig 3D). It appears that Q91 is a critical residue for substrate binding.

## GAPDH

The primary responsibility of GAPDH is to catalyze the oxidative phosphorylation of glyceraldehyde-3-phosphate to mediate the formation of ATP and NADH during glycolysis (Seidler, 2013). However, this cytoplasmic protein is also an essential metabolic regulator involved in a variety of cellular processes, including membrane fusion, transport, apoptosis, DNA replication and repair, and regulation of transcription and translation (Sirover, 1999). Besides its role in different metabolic activities, GAPDH serves as a chaperone for heme and helps maintain iron homeostasis (Boradia et al, 2014; Sweeny et al, 2018). There is strong evidence that GAPDH can interact with $\beta$-amyloid and huntingtin proteins to regulate their cytotoxicity, thus directly linking to several neurodegenerative disorders, such as Alzheimer's, Huntington's, and Parkinson's diseases (Colell et al, 2009). Pharmacologically, several anti-dementia drugs, such as tacrine, donepezil, and deprenyl, are administered to target the GAPDH apoptotic cascade process for treating dementia (Tsuchiya et al, 2004; Hara et al, 2006).

We collected a total of 8,982 single-particle projections for this class of images and identified this enzyme as GAPDH. We were able to refine its cryo-EM structure to a resolution of 3.22 Å (Figs 4 and S7 and Table S1).

Human GAPDH consists of 335 amino acids, where 333 residues are included in our final structural model. Similar to the X-ray structure of human GAPDH (PDB ID: 1U8F) (Jenkins & Tanner, 2006), the cryo-EM structure of GAPDH is tetrameric in oligomerization (Fig 4A and B). The first 149 residues of each subunit of human GAPDH consist of the N-terminal domain, which contains eight $\alpha$-helices and 11 $\beta$-stands (Fig 4C). The C-terminal domain of GAPDH constitutes residues 150–335, where these residues form six $\alpha$-helices and seven $\beta$-stands (Fig 4C).

Each subunit of GAPDH is found to possess a $NAD^+$-binding site. Most of the nucleotide-binding site is created by the N-terminal domain. However, both N- and C-terminal residues contribute to the binding. The $NAD^+$-binding site has been documented, where residues R13, I14, D35, R80, S122, and N316 are involved in forming this coenzyme-binding site (Ismail & Park, 2005; Jenkins & Tanner, 2006) (Fig 4D). Unfortunately, we did not see nucleotides occupying this $NAD^+$-binding site in our cryo-EM structure.

contribute to form the substrate-binding site are in sticks models. The positively charged arginines (R299, R319, R324, R340, and R341) that contribute to form the substrate-binding site are in magenta sticks. Residues (E134, H253, and E338) involved in binding $Mn^{2+}$ ion (green sphere) are in yellow sticks. The cryo-EM densities of these residues and bound $Mn^{2+}$ are in gray meshes.

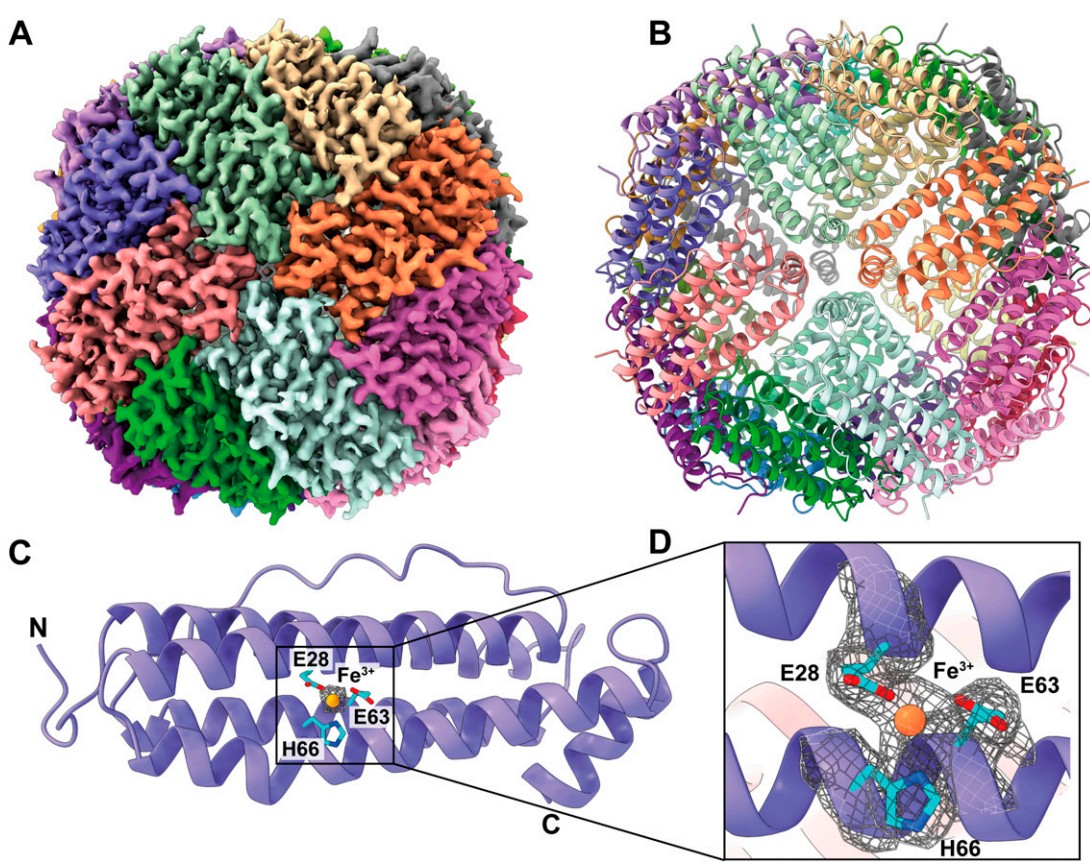

**Figure 2.   Cryo-EM structure of human FT.**
**(A)** Cryo-EM maps of human FT. The 24 subunits are colored differently. **(B)** Ribbon diagram of the structure of human FT determined by cryo-EM. In (A, B), subunits are distinguished through individual colors and match accordingly. **(C)** A single FT subunit is made up of five $\alpha$-helices. A single $Fe^{3+}$ ion (orange sphere) bound by residues E28, E63, and H66 is found in each FT monomer. These residues are in cyan sticks. The cryo-EM map of bound $Fe^{3+}$ is in gray mesh. **(D)** Zoomed view of $Fe^{3+}$ binding site. $Fe^{3+}$ is shown as an orange sphere, cryo-EM densities of bound $Fe^{3+}$ (orange sphere) and interacting residues (cyan sticks) are shown as gray meshes.

The substrate-binding site is located at the interface between the N-terminal and C-terminal domains. It has been reported that residues S151, C152, T153, T182, T211, G212, and R234 are involved in forming the catalytic site (Ismail & Park, 2005). Again, our cryo-EM structure suggests that this catalytic site is unoccupied (Fig 4E).

Interestingly, it has been observed that there is a significant inhibition of GADPH activity in the Alzheimer's disease brain related to oxidative modifications. This GADPH enzyme appears to be responsible for these modifications in the diseased brain (Butterfield et al, 2010). Residue C152 at the catalytic site is a critical residue for the activity of the enzyme. It has been shown by mutational studies in Bacillus stearothermophilus that a substitution of C149 (corresponding to C152 in human enzyme) by a serine significantly reduces GADPH activity, whereas a replacement of this cysteine with an alanine completely abolishes the enzyme activity (Didierjean et al, 1997; Boschi-Muller & Branlant, 1999).

### ALDH1A1

The ALDH1A1 enzyme plays a primary role in the biosynthesis of retinoic acid, an important signaling molecule that specifically interacts with the retinoic acid receptor. Retinoic acid signaling is a key player that controls several vital developmental processes,

including neurogenesis, cardiogenesis, and development of the eye, forelimb bud, and foregut (Duester, 2008). Mutations on this enzyme have been associated with a number of human diseases, including cancer, Parkinson's disease, and obesity (Ziouzenkova et al, 2007; Wey et al, 2012; Tomita et al, 2016). In addition, it has been observed that the expression of ALDH1A1 is significantly reduced in the Parkinson's disease brain in comparison with normal, healthy controls (Fan et al, 2021).

We collected a total of 6,684 single-particle cryo-EM projections for this class of protein images. Based on these projections, the BaR methodology allowed us to construct a high-resolution cryo-EM map. Subsequently, we were able to identify this protein as the ALDH1A1 enzyme and resolve its structure to a resolution of 3.40 Å (Figs 5 and S8 and Table S1).

ALDH1A1 assembles as a tetramer in oligomerization (Fig 5A and B). Each subunit of ALDH1A1 contains 501 amino acids. Consistent with the crystal structures of the ALDH enzymes (PDB IDs: 4WJ9 and 1BXS) (Moore et al, 1998; Morgan & Hurley, 2015), each ALDH1A1 subunit consists of a cofactor-binding domain, a catalytic domain, and an oligomerization domain (Fig 5C).

The cofactor-binding domain possesses a $NAD^+$-binding site. Presumably, residues I166, P168, W169, K193, E196, Q197, P227, F244, A231, S234, V250, S247, C303, Q350, K353, E400, and F402 are engaged

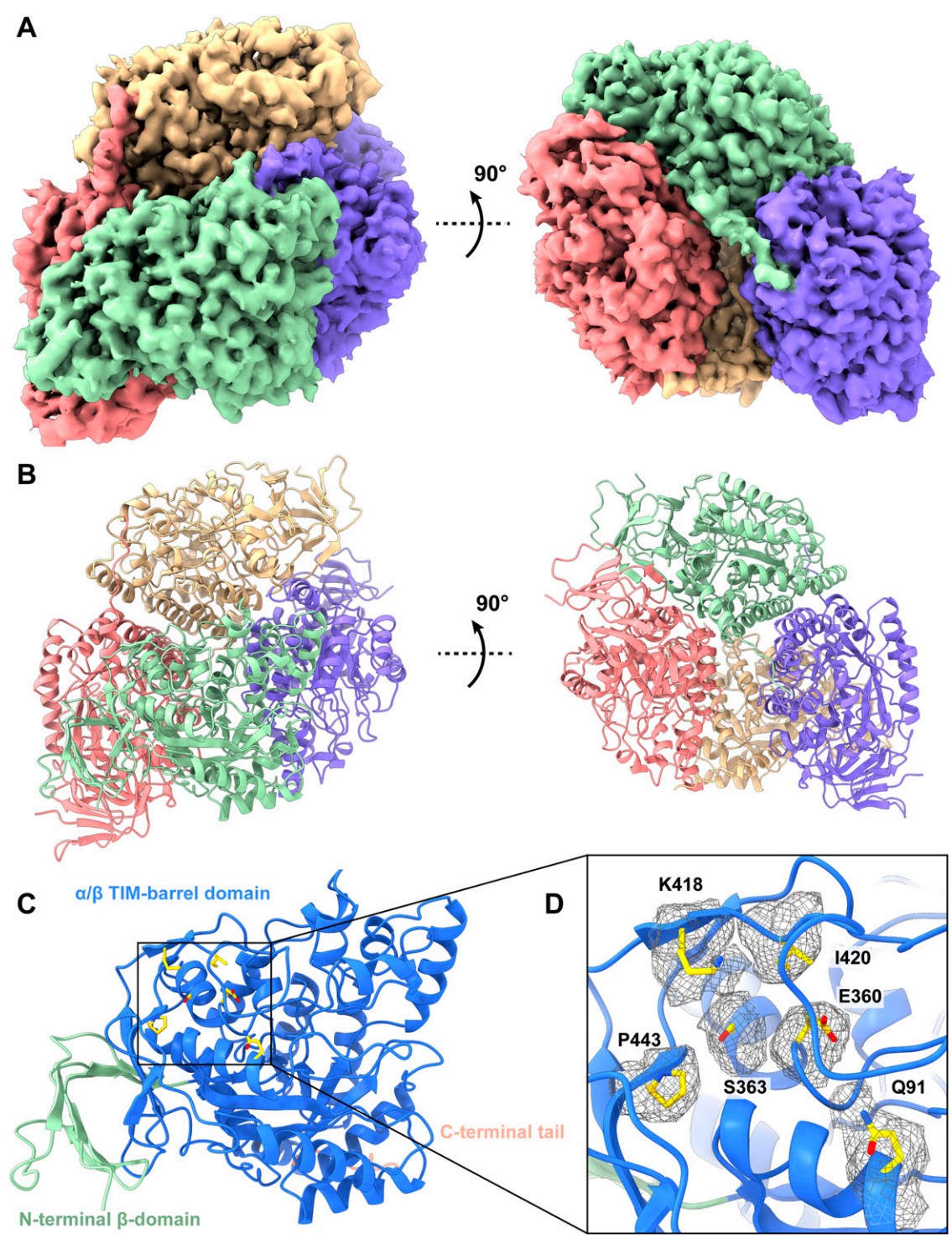

**Figure 3. Cryo-EM structure of human DPYSL2.**
**(A)** Cryo-EM maps of human DPYSL2. The four subunits of the DPYSL2 enzyme are colored differently. **(B)** Ribbon diagram of the structure of human DPYSL2 determined by cryo-EM. GAPDH forms a tetramer with D2 symmetry. In (A, B), subunits are distinguished through individual colors and match accordingly. **(C)** Structure of a human DPYSL2 subunit. Each subunit of DPYSL2 can be divided into an N-terminal β-domain (green) and a C-terminal α/β TIM-barrel domain (blue). The C-terminal unstructured tail is colored salmon. **(D)** The drug-binding site located at the C-terminal α/β TIM-barrel domain (blue). Residues Q91, E360, S363, K418, I420 and P443 involved in forming the substrate-binding site are in yellow sticks. The cryo-EM densities of these residues are in gray meshes. The C-terminal α/β TIM-barrel domain secondary structural elements are colored blue.

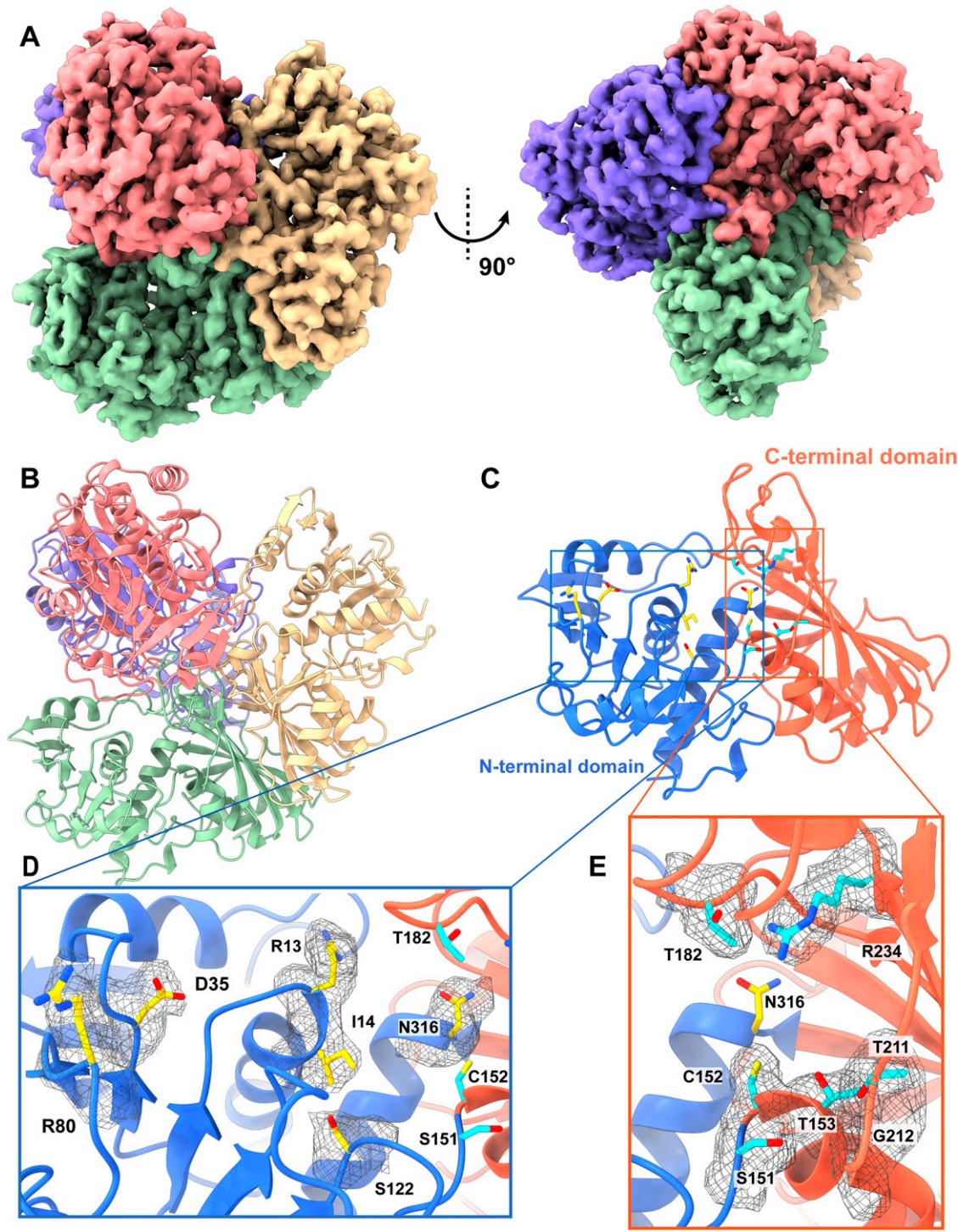

**Figure 4. Cryo-EM structure of human GAPDH.**
**(A)** Cryo-EM maps of human GAPDH. The four subunits of the GAPDH enzyme are colored differently. **(B)** Ribbon diagram of the structure of human GAPDH determined by cryo-EM. GAPDH forms a tetramer with D2 symmetry. In (A, B), subunits are distinguished through individual colors and match accordingly. **(C)** Structure of a human GAPDH subunit. Each subunit of GAPDH can be divided into N-terminal (cyan) and C-terminal (orange) domains. **(D)** The NAD$^+$-binding site located at the N-terminal domain (blue). Residues R13, I14, D35, R80, S122, and N316 involved in forming the NAD$^+$-binding site are in yellow sticks. The cryo-EM densities of these residues are in gray meshes. The C-terminal domain secondary structural elements are colored orange. **(E)** The substrate-binding site located at the interface between N-terminal (blue) and C-terminal domains (orange). Residues S151, C152, T153, T182, T211, G212, and R234 involved in forming the substrate-binding site are in cyan sticks. The cryo-EM densities of these residues are in gray meshes.

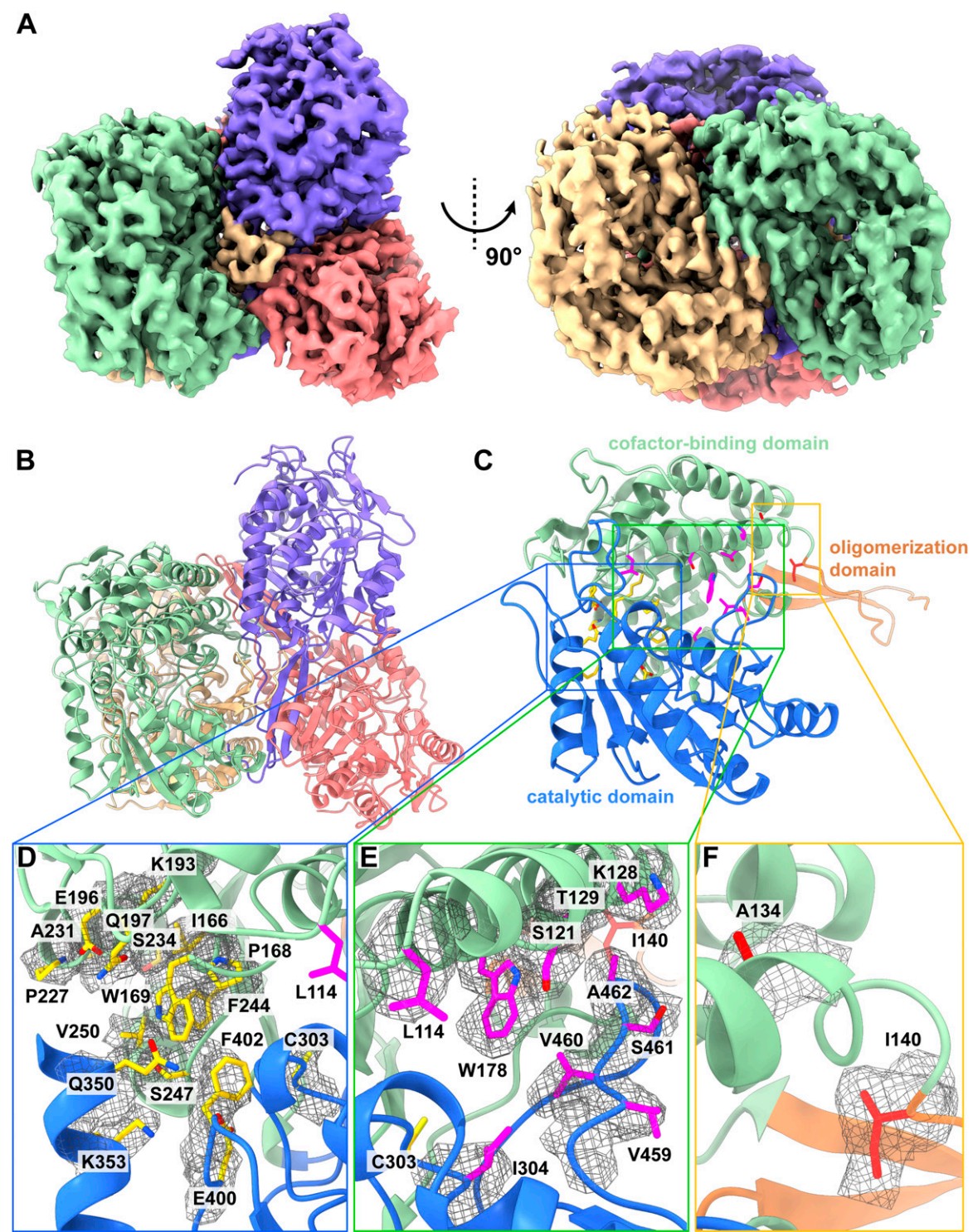

**Figure 5.  Cryo-EM structure of human ALDH1A1.**
**(A)** Cryo-EM maps of human ALDH1A1. The four subunits of the GAPDH enzyme are colored differently. **(B)** Ribbon diagram of the structure of human ALDH1A1 determined by cryo-EM. ALDH1A1 forms a tetramer with D2 symmetry. In (A, B), subunits are distinguished through individual colors and match accordingly. **(C)** Structure of a human ALDH1A1 subunit. Each subunit of ALDH1A1 can be divided into cofactor-binding (green), catalytic (blue), and oligomerization (orange) domains. **(D)** The cofactor-binding site of ALDH1A1. Residues I166, P168, W169, K193, E196, Q197, P227, F244, A231, S234, V250, S247, C303, Q350, K353, E400, and F402 are supposed to be involved in NAD$^+$ binding. These residues are in yellow sticks. The cryo-EM densities of these residues are in gray meshes. Residue L114 located within the vicinity of the cofactor-binding site is in magenta stick. The secondary structural elements of the cofactor-binding domain and catalytic domain are colored green and blue, respectively. **(E)** The substrate-binding site of ALDH1A1. Residues L114, S121, K128, T129, W178, I304, V459, V460, S461, and A462 involved in forming the substrate-binding site are in magenta sticks. The cryo-EM densities of these residues are in gray meshes. Residues I140 and C303 located within the vicinity of the substrate-binding site are in orange and

in NAD$^+$ binding (Pequerul et al, 2020) (Fig 5D). In our cryo-EM structure, we did not observe bound NAD$^+$ in our structure. The catalytic site of ALDH1A1 is surrounded by residues L114, S121, K128, T129, W178, I304, V459, V460, S461, and A462 (Pequerul et al, 2020) (Fig 5E). It has been found that this catalytic site is able to anchor ALDH1A1 inhibitors (Morgan & Hurley, 2015).

Two missense mutations, A151S and I157T, have been identified for the human ALDH1A2 enzyme (Christy & Doss, 2015), where these two mutations are strongly associated with congenital heart disease. The corresponding residues in ALDH1A1 are A134 and I140 (Fig 5F). These two residues are located at the subunit–subunit interface and are probably critical for the tetrameric oligomerization of this enzyme.

## Discussion

Biological processes of the cell are phenomenally complex events that involve networks of interactions among different proteins, biomolecules, enzymes, and even small molecules and metabolites. Therefore, the approach of systems biology is advantageous for the study of living tissues and organs, as it is capable of providing us a more comprehensive view of these biological processes. To more effectively study human tissues and organs in the context of systems proteomics, we have developed a BaR methodology to elucidate structural information of different protein components from a raw biological sample at near-atomic resolutions. We demonstrated that this methodology is able to solve structures of a number of relatively small and less abundant, unidentified proteins within a single, heterogeneous sample. It also allowed us to determine structures of both hetero-oligomeric and homo-oligomeric protein complexes in a single sample.

In the current study, we apply the BaR methodology to elucidate the composition of human brain microsomes, allowing us to demonstrate the feasibility of using a cryo-EM structural biology approach to illuminate the proteome of human tissues to high resolution. Through this structural proteomic approach, we were able to simultaneously identify and solve high-resolution cryo-EM structures of five different important enzymes from a single human brain microsomal sample at resolutions between 2.69 and 3.40 Å. The presence of these enzymes in the sample was also confirmed by proteomic study using LC–MS/MS (Table S2).

Interestingly, these five enzymes have been implicated in multiple neurodegenerative disorders, including Alzheimer's, Parkinson's, and schizophrenia diseases. For example, it has been detected that the level of GS was exceedingly high in the cerebrospinal fluid of Alzheimer's patients (Gunnersen & Haley, 1992). DPYSL2 has been indicated to be a key player in the pathogenesis of the mental illness schizophrenia (Edgar et al, 2000; Johnston-Wilson et al, 2000; Martins-de-Souza et al, 2009). Not surprisingly, FT has been directly linked to Alzheimer's and Parkinson's diseases because of the relationship between iron dyshomeostasis and

neurodegenerative disorders (Friedman et al, 2011; Ayton et al, 2015). Notably, studies have demonstrated that there is a significant inhibition of GAPDH activity in the Alzheimer's disease brain (Butterfield et al, 2010). In addition, it has been reported that there are elevated levels of aldehydes in the Parkinson's disease brain. This suggests that there is a profound loss in activity of ALDH1 enzymes in this pathologic condition as aldehydes are primarily detoxified by these enzymes (Wey et al, 2012).

There are several case reports regarding missense mutations on enzymes within the brain. (i) In two unrelated cases, newborns were found to have congenital glutamine deficiency consisting of two-point mutations (R324C and R341C) on the GS enzyme (Häberle et al, 2005). This double GS mutant causes severe brain malformations, resulting in multiorgan failure and neonatal death (Häberle et al, 2005). (ii) In the case of DPYSL2, a seizure-causing mutant Q91R has been identified, where the drug lacosamide has been found to be efficient in the reduction of symptoms (Ben-Menachem et al, 2007). It should be noted that a mutation on residue T555 of DPYSL2 to an alanine has been observed to cause the impairment of dendritic growth in cerebellar Purkinje cells (Winkler et al, 2020), although the C-terminal residues 507–572 of DPYSL2 were not included in the structural model. (iii) For the ALDH1A1 enzyme, there is strong evidence that the A134S and I140T missense mutants are strongly linked to congenital heart disease (Chen et al, 2021).

The fact that many of these enzymes are tightly associated with neurodegenerative disorders and certain cancers makes our BaR methodology exciting in that it enables us to study these proteins simultaneously in a single sample. Our work strongly indicates that BaR can be used to overcome the problem of sample impurity and heterogeneity, enabling us to use the cryo-EM structural approach to simultaneously solve structures of a variety of enzymes from tissue samples at high resolution.

## Materials and Methods

### Human brain microsome lysate

Human brain microsomes were purchased from BioIVT. The microsomes were resuspended in a buffer containing 20 mM Tris–HCl (pH 7.5), 100 mM NaCl, and 5 mM Na-cholate. The soluble lysate fraction was separated from a membrane fraction by ultracentrifugation at 20,000$g$. The extracted soluble lysate was then passed through a 0.22-$\mu$M filter and enriched using a Superdex 200 column (GE Healthcare) that was equilibrated with 20 mM Tris–HCl (pH 7.5) and 100 mM NaCl. A broad peak of protein sizes corresponding to 150–650 kD was used for cryo-EM analysis.

### Cryo-EM sample preparation and data collection

The lysate sample was concentrated to 0.67 mg/ml. 2.5 $\mu$l of sample was applied to glow discharge holey carbon grids (Cu R1.2/1.3,300

yellow sticks. The secondary structural elements of the cofactor-binding domain and catalytic domain are colored green and blue, respectively. **(F)** The oligomerization domain. Residues A134 and I140, which may be critical for oligomerization, are in red sticks. The cryo-EM densities of these residues are in gray meshes. The secondary structural elements of the cofactor-binding domain and oligomerization domain are colored green and orange, respectively.

mesh; Quantifoil) and blotted for 7 s before being plunge-frozen in liquid ethane. Data were collected using a Titan Krios cryo-electron transmission microscope (Thermo Fisher Scientific). The images were recorded using 1.0–2.5 $\mu$M defocus on a K3 direct electron detector (Gatan) using super-resolution at 81,000 × magnification. The sampling interval was 1.07 Å/pix, giving a super-resolution of 0.535 Å/pix. Micrographs were collected over 40 frames with a total dose of ~36 e$^-$/Å$^2$ in correlated double-sampling mode using serial EM (Mastronarde, 2003).

### Data processing

Data were binned by a factor of 2 and motion corrected using patch motion in cryoSPARC v3 (Punjani et al, 2017). Contrast transfer function was estimated using the patchCTF function in cryoSPARC v3. The Topaz (Bepler et al, 2019, 2020) tool, with the default ResNet16 (64 U) pre-trained model, was used to pick initial particle sets. The BaR protocol (Su et al, 2021) was used to separate the different protein structures (Fig S1). Broadly, this methodology consists of two phases. During the "build" phase, particles with structural features are selected from the preliminary 2D classes in the initial particle stack. Several rounds of 2D classification are used to further refine the selected particles. Next, potentially related subsets are sampled together and used to create initial 3D ab initio models. Models that exhibit global structural features are selected and further processed to generate starting templates. At this stage, all initial templates are built using C1 symmetry in an effort to minimize orientation bias. Initial model building is the final step in the "build" phase. During the "retrieve" phase, these initial models were used as targets for heterogeneous refinement of the initial particle stack. Using the initial particle stack allows for an increase in the particle count of each respective protein and can critically collect additional particle views that were previously obscured during the build phase. The enriched particle set for each protein was cleaned using 2D classification and refined using non-uniform refinement. Multiple iterations of the BaR procedure were performed, and the highest resolution maps were used for model building and final refinement.

### Model building and refinement

Initial models were built using previously deposited structures (Table S1) via the Uniprot database and aligned to the cryo-EM density maps in Chimera (Pettersen et al, 2004). Final models were built using Coot and refined using the phenix.real_space_refine function in phenix (Emsley & Cowtan, 2004; Adams et al, 2010). Ligands were added if there was clear density that matched previously described structural or biological data. Final structures were evaluated using MolProbity (Chen et al, 2010). Statistics associated with data collection and refinement are compiled in Table S1. Final figures were generated using the ChimeraX suite (Goddard et al, 2018).

### Proteomic analysis

4 $\mu$g of each peak of the collected brain microsome lysate was separately denatured in a buffer containing 50 mM $NH_4HCO_3$ and 8 M urea. 10 mM DTT (final concentration) was added to reduce the sample at 25°C for 30 min. The sample was then alkylated with 25 mM iodoacetamide at 25°C for 30 min. Each lysate sample was further diluted by four times using digestion buffer containing 100 mM $NH_4HCO_3$ and trypsin/Lys-C mix (1:20, enzyme:substrate). The samples were then digested overnight at 25°C. The digested peptides were desalted using a reverse-phase C18 Microspin column (Nest Group), washed twice with 150 $\mu$l aqueous solution containing 0.1% formic acid, and eluted with 150 $\mu$l aqueous solution containing 80% acetonitrile and 0.1% formic acid.

LC–MS/MS was then performed using the Thermo Scientific Fusion Lumos mass spectrometry system (Thermo Fisher Scientific). The LC column used was a Dionex 15 cm × 75 $\mu$m id Acclaim PepMap C18, 2 $\mu$m; 100 Å reversed-phase capillary chromatography column and peptides were chromatographed with a linear gradient of acetonitrile from 2% to 35% in aqueous 0.1% formic acid over 90 min at 300 nl/min. The eluent was directly introduced into the mass spectrometer operated in data-dependent MS to MS/MS switching mode with collision-induced dissociation mode. Full MS scanning was performed at 70,000 resolution between m/z of 350 and 1,500. Proteins were identified by comparing all of the experimental peptide MS/MS spectra against the UniProt human proteome database using a database search engine, MassMatrix (version 3.12). For peptide/protein identification, strict trypsin specificity was applied, the minimum peptide length was set to 6, the maximum missed cleavage was set to 2, and the cutoff false discovery rate was set to 0.025.

## Data Availability

Coordinates and EM maps are deposited under the accession codes 8DNM (PDB) & EMD-27574 (EMDB) for DPYSL2; 8DNO (PDB) & EMD-27575 for ALDH1A1; 8DNP (PDB) & EMD-27576 (EMDB) for FT; 8DNS (PDB) & EMD-27579 for GAPDH; and 8DNU (PDB) & EMD-27580 for GS.

## Supplementary Information

## Acknowledgements

We thank Dr. Philip A Klenotic for proofreading the article. This work was supported by an NIH Grant R01AI145069 (EW Yu). The mass spectrometer used was purchased with an NIH Shared Instrument Grant (S10 RR031537). We thank Belinda Willard and Ling Li for the acquisition of mass spectrometry data. We are grateful to the Cryo-Electron Microscopy Core at the CWRU School of Medicine and Dr. Kunpeng Li for access to the sample preparation and Cryo-EM instrumentation.

### Author Contributions

ML Tringides: conceptualization, data curation, formal analysis, validation, investigation, visualization, methodology, and writing—original draft, review, and editing.

Z Zhang: conceptualization, data curation, formal analysis, validation, investigation, visualization, methodology, and writing—review and editing.

CE Morgan: data curation and formal analysis.

C-C Su: data curation and formal analysis.

EW Yu: conceptualization, data curation, formal analysis, supervision, funding acquisition, validation, investigation, visualization, methodology, project administration, and writing—original draft, review, and editing.

## Conflict of Interest Statement

The authors declare that they have no conflict of interest.

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
