## [Reviewer comments · Life Science Alliance]

Life Science Alliance

A cryo-electron microscopic approach to elucidate protein structures from human brain microsomes

Marios Tringides, Zhemin Zhang, Christopher Morgan, Chih-Chia Su, and Edward Yu

DOI: <https://doi.org/10.26508/lsa.202201724>

Corresponding author(s): Edward Yu, Case Western Reserve University

Review Timeline:

Submission Date:	2022-09-13
Editorial Decision:	2022-10-13
Revision Received:	2022-10-22
Editorial Decision:	2022-11-04
Revision Received:	2022-11-07
Accepted:	2022-11-10

Transaction Report:

October 13, 2022

Re: Life Science Alliance manuscript #LSA-2022-01724

Prof. Edward W. Yu
Case Western Reserve University
Department of Pharmacology
Cleveland, Ohio 44106

Dear Dr. Yu,

Thank you for submitting your manuscript entitled "A cryo-electron microscopic approach to elucidate protein structures from human brain microsomes" to Life Science Alliance. The manuscript was assessed by expert reviewers, whose comments are appended to this letter. We invite you to submit a revised manuscript addressing the Reviewer comments.

Thank you for this interesting contribution to Life Science Alliance. We are looking forward to receiving your revised manuscript.

Sincerely,

B. MANUSCRIPT ORGANIZATION AND FORMATTING:

Reviewer #1 (Comments to the Authors (Required)):

A cryo-EM approach to elucidate protein structure from human brain microsomes
Tringides et al, 2022

Overview:

Tringides et al present a concise cryo-EM study identifying and visualizing 5 proteins from brain microsomes; Glutamine synthetase, Ferritin, DPYSL2, GAPDH, and ALDH1A1 using the previously-described cryo-EM workflow of Build-and-Retrieve, which includes preparation of cell lysate, protein isolation by SEC, cryo-EM analysis and model building. This work is a testament to the ever-more challenging nature of targets of cryo-EM structural characterization and represents the trajectory of not just visualizing, but also identifying critical cellular complexes and elucidating their functions by cryo-EM.

Major comments

This brief and descriptive study relies heavily on work presented by Su et al in Nature Methods. As a result, the manuscript is highly derivative in its nature. Due to the technical aspects of this study, more in-depth methodological details should be included (rather than referenced to in the Su et al).

On page 5, it is stated that several rounds of 2D classification were performed to allow the authors to sort images into different protein classes, but the starting material (initial classes) are not presented in S1 and it is thus difficult to assess the complexity of the method and how robust the retrieval is. Initial versus final 2D should be shown to illustrate particle heterogeneity and provide further insight into the BnR process.

It is unclear what sample was analyzed for EM, i.e., what part(s) or the very broad SEC peak and to what degree SEC contributed to "sample purification".

It's unclear to me how particle picking was performed; where all particles in each sample (containing particles ranging in size) picked on were parameters used to only pick certain particles?

Reviewer #2 (Comments to the Authors (Required)):

This manuscript by Tringides et al. describes the structural determination of five different brain enzymes from a raw brain microsomal lysate using the "Build and Retrieve (BaR)" workflow of cryo-EM reported by the same group previously, combined with mass spectrometry. This is an informative study and provides new strategy for the community to elucidate protein structures from a heterogeneous protein sample. I have only a couple of questions/comments for the authors to address:

1) It's unclear which fraction of the size exclusion chromatography applied to the cryo-EM sample preparation. It should be shown and labeled in the supplementary figure.

2) The authors should include the PDB ID of previous determined X-ray structures when they compare them with the structures solved in present study.

3) The authors state that C-terminal tail of DPYSL2 is readily cleaved off both in vivo and in vitro, why we can still observe it in the structure?

4) On page 9: the authors state that "The large α/β TIM-barrel domain possess a substrate-binding site. It has been observed that residues D80, Q91, E111, Y167 and R173 are engaged in substrate binding". If the substrate binding site was reported by the authors, please share the biochemical evidence; if it was reported by other group, please add the reference. Similarly, please add reference on page 11 when you mention that "It has been reported that residues S151, C152, T153, T182, T211, G212 and R234 involved in forming the catalytic site."

5) Typo: page 6, "Within the active site of the C-terminal catalytic domain of each subunit of human CS" and Page 19, Figure 1 legends: "CS is organized into a". It should be "GS".

We thank the reviewers for their thoughtful and productive comments. Listed below are the original comments with corresponding responses.

Reviewer #1 (Comments to the Authors (Required)):

A cryo-EM approach to elucidate protein structure from human brain microsomes
Tringides et al, 2022

Overview:

Tringides et al present a concise cryo-EM study identifying and visualizing 5 proteins from brain microsomes; Glutamine synthetase, Ferritin, DPYSL2, GAPDH, and ALDH1A1 using the previously-described cryo-EM workflow of Build-and-Retrieve, which includes preparation of cell lysate, protein isolation by SEC, cryo-EM analysis and model building. This work is a testament to the ever-more challenging nature of targets of cryo-EM structural characterization and represents the trajectory of not just visualizing, but also identifying critical cellular complexes and elucidating their functions by cryo-EM.

We very much appreciate this reviewer's comment, highlighting that "this work is a testament to the ever-more challenging nature of targets of cryo-EM structural characterization and represents the trajectory of not just visualizing, but also identifying critical cellular complexes and elucidating their functions by cryo-EM".

Major comments

This brief and descriptive study relies heavily on work presented by Su et al in Nature Methods. As a result, the manuscript is highly derivative in its nature. Due to the technical aspects of this study, more in-depth methodological details should be included (rather than referenced to in the Su et al).

The Build-and-Retrieve procedures have been rewritten in the "Method" section of this revised manuscript to describe the details of the protocol, following the reviewer's comment.

Additional technical details have also been added. For example, in an effort to more clearly explain the BaR methodology, a new paragraph "Briefly, several 2D classification iterative rounds permitted us to sort the images into different protein classes (Figures S2 and S3). ... cryo-EM map construction and final structural refinement." (p. 5) has been included in this revised version of the manuscript.

On page 5, it is stated that several rounds of 2D classification were performed to allow the authors to sort images into different protein classes, but the starting material (initial classes) are not presented in S1 and it is thus difficult to assess the complexity of the method and how robust the retrieval is. Initial versus final 2D should be shown to illustrate particle heterogeneity and provide further insight into the BnR process.

An additional supplementary figure (Figure S3) has been included in this revised version of the manuscript to address this. In this figure, the workflow for the GAPDH enzyme was included.

Initial 2D views are shown as well as “middle set” of processed particles used to build the initial templates. The final particles and model are also included to demonstrate the potential for BaR to pick up novel and higher resolution particle views.

It is unclear what sample was analyzed for EM, i.e., what part(s) or the very broad SEC peak and to what degree SEC contributed to "sample purification".

Our goal for using SEC is to enrich protein particles, allowing us to obtain more particle counts. We only focus on particles with size ≥ 100 kDa, which would make it easier for high-resolution cryo-EM structural determination. During the enrichment process, we were also able to separate the large peak corresponding to aggregation from protein particles. For size ≥ 100 kDa, we only observed two peaks, 100-300 kDa and 300-500 kDa, from our sample. We therefore separately analyzed these two protein peaks using cryo-EM and mass spectrometry. A new supplementary figure (Figure S1) has been included in this revised manuscript to more clearly show the two peaks used for our experiments.

It's unclear to me how particle picking was performed; where all particles in each sample (containing particles ranging in size) picked on were parameters used to only pick certain particles?

The Topaz (Bepler *et al.*, 2019, 2020) tool, with the default ResNet16 (64 units) pretrained model, was used to pick initial particle sets. Additional technical details have been provided in the main text (p. 5). We have also rewritten the “Data processing” section on p. 17 of “Materials and Methods” of this revised manuscript. In addition, a new supplementary figure (Figure S3) has also been added to better illustrate the workflow using the GAPDH model building as an example.

Reviewer #2 (Comments to the Authors (Required)):

This manuscript by Tringides et al. describes the structural determination of five different brain enzymes from a raw brain microsomal lysate using the "Build and Retrieve (BaR)" workflow of cryo-EM reported by the same group previously, combined with mass spectrometry. This is an informative study and provides new strategy for the community to elucidate protein structures from a heterogeneous protein sample. I have only a couple of questions/comments for the authors to address:

We thank this reviewer’s glowing comment, indicating that “this is an informative study and provides new strategy for the community to elucidate protein structures from a heterogeneous protein sample”.

1) It's unclear which fraction of the size exclusion chromatography applied to the cryo-EM sample preparation. It should be shown and labeled in the supplementary figure.

A new supplementary figure (Figure S1), showing the two SEC peaks, has been added in this revised version of the manuscript, following this reviewer’s suggestion.

2)The authors should include the PDB ID of previous determined X-ray structures when they compare them with the structures solved in present study.

The PDB IDs (2UU7 for GS, 2FHA for FT, 1KCX and 5MKV for DPYSL2, 1U8F for GAPDH, and 4WJ9 and 1BXS for ALDH1A1) have been included in this revised manuscript, following this reviewer's suggestion.

3)The authors state that C-terminal tail of DPYSL2 is readily cleaved off both in vivo and in vitro, why we can still observe it in the structure?

We actually could only observe a portion (residues 488-496) of the C-terminal tail of DPYSL2 with well-defined cryo-EM densities. The remainder of the tail (residues 497-506) exhibited very weak densities, indicating that this portion of the tail may be very flexible, or this tail may be truncated in some protein particles. We therefore decided to remove residues 497-506 of this C-terminal tail to redraw Figure 3 in this revised manuscript. After the removal of these tail residues (497-506), we observed that the remaining C-terminal tail of each DPYSL2 subunit still contacts its neighboring subunit (see revised Figure 3). In addition, the statements on p. 10 of this revised manuscript has been modified to "Additionally, the C-terminal tail has been characterized as a site of proteolytic cleavage. Unexpectedly, we observed relatively strong cryo-EM densities for a portion of this C-terminus tail (residues 488-496). The structure indicates that each subunit of this C-terminal tail (residues 488-496) appears to cross over to the next subunit, directly contacting the next subunit to enhance subunit-subunit interactions."

4)On page 9: the authors state that "The large α/β TIM-barrel domain possess a substrate-binding site. It has been observed that residues D80, Q91, E111, Y167 and R173 are engaged in substrate binding". If the substrate binding site was reported by the authors, please share the biochemical evidence; if it was reported by other group, please add the reference. Similarly, please add reference on page 11 when you mention that "It has been reported that residues S151, C152, T153, T182, T211, G212 and R234 involved in forming the catalytic site."

Appropriate references have added in this revised version of the manuscript, following this reviewer's suggestion. For example, citations has been added at the end of the statements "... observed that residues D80, Q91, E111, Y167 and R173 are engaged in substrate binding (Moutal *et al.*, 2019)" (p. 10), and "... residues S151, C152, T153, T182, T211, G212 and R234 involved in forming the catalytic site (Ismail & Park, 2005)" (p. 12) of this revised manuscript.

5)Typo: page6, "Within the active site of the C-terminal catalytic domain of each subunit of human CS" and Page 19, Figure1 legends: "CS is organized into a". It should be "GS".

We thank the reviewer for catching these typos. The typos have been fixed in this revised version of the manuscript.

November 4, 2022

RE: Life Science Alliance Manuscript #LSA-2022-01724R

Prof. Edward W. Yu
Case Western Reserve University
Department of Pharmacology
Cleveland, Ohio 44106

Dear Dr. Yu,

Thank you for submitting your revised manuscript entitled "A cryo-electron microscopic approach to elucidate protein structures from human brain microsomes". We would be happy to publish your paper in Life Science Alliance pending final revisions necessary to meet our formatting guidelines.

- please upload your supplementary figures as single files
- please add the Twitter handle of your host institute/organization as well as your own or/and one of the authors in our system
- please add the author contributions to the main manuscript text
- please add the supplementary figure legends to the main manuscript text
- please add a conflict of interest statement to your main manuscript text
- the pdb accession numbers and EMDB accession codes should now be made publicly accessible

Figure Check:

-for Figure 1, panels E and F are mentioned in the figure legend, but these are not in the figure, and there aren't any callouts for these panels in the main text

A. FINAL FILES:

B. MANUSCRIPT ORGANIZATION AND FORMATTING:

Sincerely,

Reviewer #1 (Comments to the Authors (Required)):

I am content with the author's responses to my comments.

Reviewer #2 (Comments to the Authors (Required)):

I am satisfied with the revised version, and I recommend publication in the journal.

Figure Check:

-for Figure 1, panels E and F are mentioned in the figure legend, but these are not in the figure, and there aren't any callouts for these panels in the main text

Yes, Figure 1 only have panels A to D. The typos have been fixed in this revised manuscript.

In addition, we have requested to release all PDBs and EMDBs and made them available to the public.

November 10, 2022

RE: Life Science Alliance Manuscript #LSA-2022-01724RR

Prof. Edward W. Yu
Case Western Reserve University
Department of Pharmacology
Cleveland, Ohio 44106

Dear Dr. Yu,

Thank you for submitting your Research Article entitled "A cryo-electron microscopic approach to elucidate protein structures from human brain microsomes". It is a pleasure to let you know that your manuscript is now accepted for publication in Life Science Alliance. Congratulations on this interesting work.

DISTRIBUTION OF MATERIALS:

Again, congratulations on a very nice paper. I hope you found the review process to be constructive and are pleased with how the manuscript was handled editorially. We look forward to future exciting submissions from your lab.

Sincerely,
